# Experiences of Dutch maternity care professionals during the first wave of COVID-19 in a community based maternity care system

Eline L. M. van Manen[1]*, Martine Hollander[1], Esther Feijen-de Jong[2,3], Ank de Jonge[2], Corine Verhoeven[2,4,5], Janneke Gitsels[2]

1 Department of Obstetrics, Amalia Children's Hospital, Radboud University Medical Center, Nijmegen, The Netherlands, 2 Midwifery Science, AVAG (Academy Midwifery Amsterdam and Groningen), Amsterdam Public Health Research Institute, Amsterdam University Medical Center, Vrije Universiteit Amsterdam, Amsterdam, The Netherlands, 3 Department of General Practice & Elderly Medicine, AVAG (Academy Midwifery Amsterdam and Groningen), University Medical Center Groningen, University of Groningen, Groningen, The Netherlands, 4 Department of Obstetrics and Gynecology, Maxima Medical Centre, Veldhoven, The Netherlands, 5 Division of Midwifery, School of Health Sciences, University of Nottingham, Nottingham, United Kingdom

* eline.vanmanen@radboudumc.nl

**Data Availability Statement:** Data are available from the DANS database at DOI: https://doi.org/10.17026/dans-zjh-9ae3.

## Abstract

### Background and objective

During the COVID-19 pandemic the organization of maternity care changed drastically; this study into the experiences of maternity care professionals with these changes provides suggestions for the organization of care during and after pandemics.

### Design

An online survey among Dutch midwives, obstetricians and obstetric residents. Multinomial logistic regression analyses were used to investigate associations between the respondents' characteristics and answers.

### Results

Reported advantages of the changes were fewer prenatal and postpartum consultations (50.1%). The necessity and safety of medical interventions and ultrasounds were considered more critically (75.9%); 14.8% of community midwives stated they referred fewer women to the hospital for decreased fetal movements, whereas 64.2% of the respondents working in hospital-based care experienced fewer consultations for this indication. Respondents felt that women had more confidence in giving birth at home (57.5%). Homebirths seemed to have increased according to 38.5% of the community midwives and 65.3% of the respondents working in hospital-based care. Respondents appreciated the shift to more digital consultations rather than face-to-face consultations. Mentioned disadvantages were that women had appointments alone, (71.1%) and that the community midwife was not allowed to join a woman to obstetric-led care during labour and subsequently stay with her (56.8%).

**Funding:** The author(s) received no specific funding for this work.

**Competing interests:** The authors have declared that no competing interests exist.

Fewer postpartum visits by family and friends led to more tranquility (59.8%). Overall, however, 48.0% of the respondents felt that the safety of maternity care was compromised due to policy changes.

## Conclusions

Maternity care professionals were positive about the decrease in routine care and the increased confidence of women in home birth, but also felt that safety in maternity care was sometimes compromised. According to the respondents in a future crisis situation it should be possible for community midwives to continue to deliver a personal handover after the referral of women to the hospital, and to stay with them.

## Introduction

At the end of February 2020, the first cases of the Coronavirus disease (COVID-19) appeared in the Netherlands. At first, the impact of COVID-19 seemed limited, but the number of cases increased rapidly during March 2020. Measures were taken to decrease the risk of infection [1]. As the number of COVID-19 infected patients increased, routine medical care was scaled down to the minimum care necessary [2]. Also, in maternity care face-to-face contact between women and caregivers was minimized, which impacted the organization and utilization of maternity care profoundly.

The organization of maternity care in the Netherlands differs from most other countries. Low-risk pregnant women are cared for by a community midwife, within community-based healthcare. Low-risk women at the onset of labour are attended by their community midwife and have the choice to give birth at home (12.7% of all births in 2019) or in a birth centre or as an outpatient in a hospital (14.6% of all births in 2019) [3]. If problems arise, women are referred to a hospital for prenatal care or during birth [3]. After birth in the hospital, women usually stay there for a short period of time. Women and their babies are usually supported at home by a community midwife and a maternity care assistant. The maternity care assistant assists the mother with her baby and carries out light domestic work during the first eight days, for approximately six hours per day. For a more detailed description of the Dutch maternity care system, see Perdok et al [4].

The Royal Dutch Organization of Midwives (KNOV) published a schedule for prenatal care in 2008, stating that a term pregnancy on average consists of thirteen prenatal consultations, and two prenatal ultrasounds are offered routinely [5]. This number is higher than the number of prenatal visits recommended by the World Health Organisation (WHO) and international guidelines, but is based on the fact that women appreciate regular contact for support and information during pregnancy [6, 7]. At the start of the pandemic, the KNOV, the Dutch Society of Obstetrics and Gynaecology (NVOG) and the professional organization for maternity care assistants provided a guideline to minimize maternity care in the Netherlands during the COVID-19 pandemic [8]. Prenatal care was reduced to seven consultations for a term pregnancy, while two prenatal ultrasounds continued to be offered [8, 9]. Clients were called prior to a face-to-face consultation, to explain the measures that were being taken regarding health and safety, to triage for COVID-19 related symptoms and to discuss the social and medical situation of the client. They were advised to come to appointments alone [8]. These regulations ensured that the number of consultations was cut by half and carried out by phone or video call as much as possible.

Similar changes were instituted for postnatal care. In 2018, the KNOV advised four postnatal home visits in the first ten days after birth [10]. At the start of the COVID-19 pandemic, home visits were minimized; they were only recommended for medical or psychosocial reasons. All other contacts were through (video)calling or so-called 'window visits'. During window visits, the community midwife observes the mother and her baby from outside, behind the window [8]. Therefore, most women did not have any in-person home visits by their community midwife. Postnatal care by maternity care assistants remained as it was before.

In addition to the reduction in appointments, there were additional changes in hospitals. During labour, only the partner (or one other person) was allowed to be present. In some hospitals water immersion or a water birth was (temporarily) no longer allowed [11]. The community midwife was not allowed to stay during labour in case she referred the woman to the obstetrician because of the occurrence of complications, and giving birth at the hospital with a community midwife was no longer an option in a few hospitals.

The precautions were not merely to protect pregnant women, but also to protect the maternity care professionals; in the Netherlands, at the start of June 2020, more than a third of registered infections were among health care professionals. However, until the 1st of June no tests were available to the general public; therefore, health care professionals were tested more often than the general population [1].

Knowledge of the impact of COVID-19 on pregnancy and labour is increasing. According to a meta-analysis by Allotey et al. (2020), pregnant women are more likely to need admission to an intensive care unit compared with non-pregnant women. In pregnant women infected with COVID-19, there seems to be an increased risk of caesarean section and preterm birth compared to pregnant women that are not infected [12]. There is limited evidence for vertical transmission [13–16]. There may have been an overall decrease in preterm births during the COVID-19 lockdown period [17, 18]. To date, there are no studies on how maternity care professionals in the Netherlands experienced the organizational changes in maternity care during the pandemic. Researching their opinions may not only support current changes in the structure of maternity care, but also during possible future crises.

This article focuses on the following research question: What are the opinions and experiences of maternity care professionals with the organization of maternity care during the COVID-19 pandemic? And what are opportunities for the long term organization of maternity care?

## Methods

### Study design

This survey using digital questionnaires was part of a larger study (the WAAG-study). The WAAG-study is a mixed-methods study evaluating the consequences of the COVID-19 pandemic on the organization of maternity care in the Netherlands, by examining the experiences of maternity care professionals, pregnant and postpartum women, their partners and other stakeholders. Ethical approval was not deemed necessary by the Medical Ethics Committee of the University of Amsterdam (METc), because participants received no medical interventions, and the emotional burden of the questions was not considered to be so severe that approval of a medical ethical committee was warranted (2020.255).

### Respondents

Maternity care professionals were eligible for participation if they were an obstetrician, resident in obstetrics, community- or hospital-based midwife and actively working in maternity care during the COVID-19 pandemic in the Netherlands.

## Measurement tool

A questionnaire was designed specifically for this study and made available via Survalyzer. The questionnaire was online for four weeks from the 30th of May until the 29th of June 2020. It consisted of 28 questions (See S1 Questionnaire). Seven questions were about respondents' characteristics. Four questions concerned advantages and disadvantages of the changes in maternity care due to COVID-19. For these questions, multiple answer options were provided, and the respondents could give a maximum of three answers. Four questions concerned the topics "measures during COVID-19", "cooperation within the maternity care collaboration (Verloskundig Samenwerkingsverband, VSV)", "capacity in the maternity wards and neonatal departments", and "transfers between levels of care". Three questions consisted of a five-point Likert scale, concerning the topics "safety of maternity care", "job satisfaction" and "policy of personal protective equipment (PPE)". Six questions asked about the effect of COVID-19 on different organizational policies such as choice in place of birth and consultations for fetal movement.

We invited respondents to participate through social media (Twitter and Facebook) and professional organizations (the NVOG, the society for residents in gynaecology and obstetrics (VAGO) and the KNOV) through newsletters and direct mailing. The invitations distributed through social media and the professional organizations directed the respondents to a website, exclusively designed for this study (www.coronageboortezorg.nl). Informed consent was given by filling out the questionnaire.

## Analysis

The data were imported from Survalyzer and analyzed using IBM SPSS Statistics for Windows, version 26 (IBM Corporation Inc., Armonk, NY, USA). Participants only providing background information were excluded; if a respondent partially filled out the questionnaire, only the provided data were analyzed. Most questions included a free text field marked as 'other'. Results from free text fields were either recoded into existing categories or new categories. Questions with a five-point Likert scale were recoded into three categories ('yes', 'neutral' and 'no'). The characteristics "working region" and whether the respondent was working in a municipality severely affected by COVID-19 or not, were established using the postal code of the community-based working address of the respondent. The working regions were divided into four regions: North-Holland, East-Holland, South-Holland, and West-Holland, as stated by the Nomenclature des Unités Territoriales Statistiques 1 (NUTS 1) [19]. Severely affected municipalities were defined as having more than 495 infections with COVID-19 per 100.000 inhabitants, as this was the cut-off point used by the National Institute for Public Health and Environment (RIVM) [20]. Baseline characteristics (profession, age, gender, years of work experience, working region and infection with COVID-19) were analyzed using descriptive statistics. Chi-square tests were used to find possible associations between community-based care and hospital-based care. Binominal and multinomial linear regression analyses were performed to analyze a possible relationship between the characteristics "profession", "age", "work experience", "working region" and whether the respondent was working in a severely affected municipality and the provided answers. Odds-ratios (OR) were calculated with a corresponding 95% confidence interval (95% CI); p-values <0.05 were considered significant. Only the answer categories mentioned by at least ten percent of the respondents are shown in the tables. Remaining answers were recoded into 'other'.

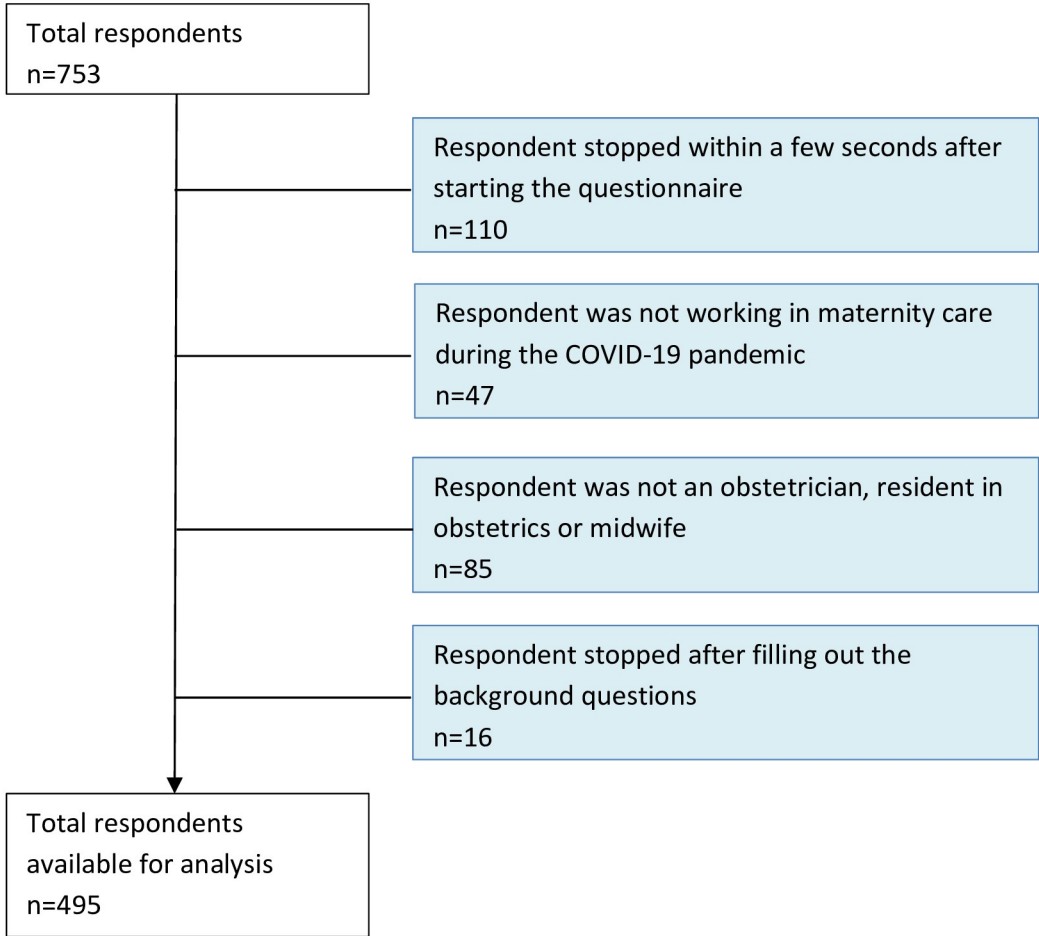

**Fig 1. Flowchart of the study population.**

## Results

The total number of respondents was 753. After exclusion for stopping with the questionnaire early or not meeting the inclusion criteria (n = 258), 495 respondents were available for analysis (Fig 1). The characteristics of the respondents are shown in Table 1. Three quarters (364, 73.5%) of the respondents were community midwives.

### Impact of COVID-19 on prenatal care

Table 2 demonstrates the advantages and disadvantages of the changes in prenatal care as reported by the respondents. The most important advantage was more deliberation about the necessity and safety of medical interventions and ultrasounds (75.9%). Of all respondents, 259 (56.7%) found fewer prenatal consultations an advantage; this was mentioned more often by respondents working in hospital-based care than in community-based care (68.8% vs. 52.5%, P<0.01). Respondents working in hospital-based care more often had a positive experience with telephone- or video consultations than respondents working in community-based care (26.3% vs. 14.7%, P = 0.01).

The most significant disadvantage maternity care providers experienced was that women had consultations and ultrasounds alone, without being companied by their partner, family or

**Table 1. Characteristics of the study population.**

| Characteristics | N | (%) |
|---|---|---|
| **Total** | 495 | (100) |
| **Gender** | | |
| Male | 12 | (2.4) |
| Female | 481 | (97.6) |
| Missing | 2 | |
| **Age (years)** | | |
| ≤30 | 120 | (24.3) |
| 31–40 | 153 | (30.9) |
| 41–50 | 136 | (27.5) |
| 51–60 | 66 | (13.3) |
| >60 | 20 | (4.0) |
| **Profession** | | |
| Community midwife | 364 | (73.5) |
| Hospital-based midwife | 75 | (15.2) |
| Obstetrician | 34 | (6.9) |
| Resident obstetrics | 22 | (4.4) |
| **Work experience (years)** | | |
| ≤5 | 105 | (21.2) |
| 6–10 | 90 | (18.2) |
| 11–15 | 88 | (17.8) |
| 16–20 | 89 | (18.0) |
| >20 | 123 | (24.8) |
| **Working region** | | |
| North-Netherlands | 53 | (10.7) |
| East-Netherlands | 126 | (25.5) |
| South-Netherlands | 107 | (21.6) |
| West-Netherlands | 209 | (42.2) |
| **Working in a municipality severely affected by COVID-19** | | |
| Yes | 28 | (5.7) |
| No | 467 | (94.3) |
| **Infection with COVID-19** | | |
| Tested positive for COVID-19 | 7 | (1.7) |
| COVID-19 symptoms, not tested | 31 | (7.5) |
| No symptoms of COVID-19 | 375 | (90.8) |
| Missing | 82 | |

a friend (71.1%). A second disadvantage mentioned by 53.0% of the respondents, more often by respondents working in community-based care compared to hospital-based care, was that they felt that a decrease in prenatal consultations caused more uncertainty for women (58.4% vs. 37.3%, P<0.01). Third, 152 (33.3%) of the respondents mentioned as a disadvantage the fact that women did not want to go to the hospital or midwifery practice, because of fear of getting infected. This was more often mentioned by respondents working in hospital-based care than respondents working in community-based care (66.1% vs 21.8%, P<0.01).

## Impact of COVID-19 on intrapartum care

Table 3 shows the advantages and disadvantages of the changes in intrapartum care mentioned by the respondents. The most frequently mentioned advantage was the impression of the

**Table 2. Advantages and disadvantages of the changes in prenatal care as experienced by maternity care professionals.**

| Advantages | Total (n = 457) | % | A: Community midwife (n = 339) | B: Hospital-based midwife (n = 64) | C: Obstetrican (n = 34) | D: Resident obstetrics (n = 20) | P-value between A and B+C+D |
|---|---|---|---|---|---|---|---|
| More deliberation about the necessity and safety of ultrasounds and medical interventions | 347 | 75.9% | 247 (72.9%) | 55 (85.9%) | 27 (79.4%) | 18 (90.0%) | 0.01* |
| Fewer prenatal consultations | 259 | 56.7% | 178 (52.5%) | 42 (65.6%) | 20 (58.8%) | 19 (95.0%) | <0.01* |
| Positive experiences with telephone or video consultations | 81 | 17.7% | 50 (14.7%) | 13 (20.3%) | 12 (35.3%) | 6 (30.0%) | 0.01* |
| Not shaking hands anymore | 58 | 12.7% | 35 (10.3%) | 12 (18.8%) | 7 (20.6%) | 4 (20.0%) | 0.02* |
| Collaboration between community-based and hospital-based care improved | 51 | 11.2% | 44 (13.0%) | 1 (1.6%) | 6 (17.6%) | 0 (0.0%) | 0.04* |
| Other | 84 | 18.4% | 50 (14.7%) | 22 (34.4%) | 9 (26.5%) | 3 (15.0%) | |
| None | 45 | 9.8% | 42 (12.4%) | 2 (3.1%) | 1 (2.9%) | 0 (0.0%) | <0.01* |
| **Disadvantages** | | | | | | | |
| Women had to come to consultations and ultrasounds alone(without their partner) | 325 | 71.1% | 247 (72.9%) | 43 (67.2%) | 23 (67.6%) | 12 (60.0%) | 0.19 |
| Decrease in prenatal consultations caused more uncertainty for women | 242 | 53.0% | 198 (58.4%) | 24 (37.5%) | 14 (41.2%) | 6 (30.0%) | <0.01* |
| Women did not want to go to the midwifery practice or hospital, afraid of getting infected with COVID-19 | 152 | 33.3% | 74 (21.8%) | 44 (68.8%) | 22 (64.7%) | 12 (60.0%) | <0.01* |
| A decrease in ultrasounds caused more uncertainty for women | 125 | 27.4% | 116 (34.2%) | 4 (6.3%) | 1 (2.9%) | 4 (20.0%) | <0.01* |
| Women were reluctant to call the practice or hospital, afraid to be a burden | 113 | 24.7% | 72 (21.2%) | 22 (34.4%) | 8 (23.5%) | 11 (55.0%) | <0.01* |
| Other | 161 | 35.2% | 134 (39.5%) | 14 (21.9%) | 9 (26.5%) | 4 (20.0%) | |
| None | 8 | 1.8% | 6 (1.8%) | 0 (0.0%) | 2 (5.9%) | 0 (0.0%) | 0.96 |

Respondents were allowed to give up to three answers, therefore the total can be higher than the total number of respondents

* P<0.05

respondents that women and partners were less afraid to give birth at home (57.5%), more often mentioned by respondents working in community-based care than hospital-based care (72.2% vs. 12.8%, P<0.01). An advantage stated by 32.5% of the respondents, more often by respondents working in hospital-based care than community-based care, was that fewer people were present during labour (54.1% vs. 25.4%, P<0.01). Of all respondents, 107 (24.3%) stated there were fewer unnecessary admissions to the hospital (35.8% hospital-based care vs. 20.5% community-based care, P<0.01). A disadvantage reported by 56.8% of the respondents, more often by respondents working in community-based care than in hospital-based care, was that the community midwife was not allowed to handover in person in case of referral and stay with the woman during labour (64.7% vs. 33.0%, P<0.01). Forty percent of all respondents stated that they were afraid for their own safety or their families, because keeping a safe distance during labour was almost impossible.

## Opportunities for future organization of maternity care

The respondents were asked to name opportunities for the organization of maternity care in the future (Table 4). The measure mentioned most often was a decrease in consultations and ultrasounds when there is no medical indication (50.1%). This was stated more often by respondents working in hospital-based care than community-based care (63.4% vs. 46.0%,

**Table 3. Advantages and disadvantages of the changes in intrapartum care as experienced by maternity care professionals.**

| Advantages | Total (n = 440) | % | A: Community midwife (n = 331) | B: Hospital-based midwife (n = 60) | C: Gynaecologist (n = 31) | D: Resident obstetrics (n = 18) | P-value between A and B+C+D |
|---|---|---|---|---|---|---|---|
| Women and partners were less scared to give birth at home (either planned or unplanned) | 253 | 57.5% | 239 (72.2%) | 8 (13.3%) | 3 (9.7%) | 3 (16.7%) | <0.01* |
| Fewer people were present during labour | 143 | 32.5% | 84 (25.4%) | 34 (56.7%) | 18 (58.1%) | 7 (38.9%) | <0.01* |
| Fewer unnecessary admissions to hospital | 107 | 24.3% | 68 (20.5%) | 26 (43.3%) | 10 (32.3%) | 3 (16.7%) | <0.01* |
| Fewer capacity problems | 71 | 16.1% | 52 (15.7%) | 11 (18.3%) | 5 (16.1%) | 3 (16.7%) | 0.66 |
| No medical or midwifery students present during labour | 61 | 13.8% | 31 (9.4%) | 22 (36.1%) | 3 (9.7%) | 5 (27.8%) | <0.01* |
| Collaboration between community-based and hospital-based care improved | 46 | 10.5% | 39 (11.8%) | 1 (1.7%) | 5 (16.1%) | 1 (5.6%) | 0.15 |
| Other | 84 | 19.1% | 55 (16.6%) | 22 (36.7%) | 6 (19.4%) | 1 (5.6%) | |
| None | 50 | 11.4% | 35 (10.6%) | 5 (8.3%) | 5 (16.1%) | 5 (27.8%) | 0.39 |
| **Disadvantages** | | | | | | | |
| The community midwife was not allowed to deliver a personal handover and stay with the woman in labour after referral | 250 | 56.8% | 214 (64.7%) | 22 (36.7%) | 5 (16.1%) | 9 (50.0%) | <0.01* |
| It was impossible to keep a safe distance from other people, so I was afraid for my own safety or for the safety of my family | 174 | 39.5% | 137 (41.4%) | 26 (43.3%) | 9 (29.0%) | 2 (11.1%) | 0.18 |
| Water birth was not allowed | 128 | 29.1% | 101 (30.5%) | 19 (31.7%) | 6 (19.4%) | 2 (11.1%) | 0.28 |
| Fewer people were present during labour | 100 | 22.7% | 85 (25.7%) | 8 (13.3%) | 5 (16.1%) | 2 (11.1%) | 0.01* |
| Less contact with women because of the use of PPE | 95 | 21.6% | 48 (14.5%) | 26 (43.3%) | 13 (41.9%) | 8 (44.4%) | <0.01* |
| No medical or midwifery students present during labour | 54 | 12.3% | 41 (12.4%) | 6 (10.0%) | 7 (22.6%) | 0 (0.0%) | 1.00 |
| Other | 174 | 39.5% | 115 (34.7%) | 30 (50.0%) | 14 (45.2%) | 15 (83.3%) | |
| None | 22 | 5.0% | 14 (4.2%) | 3 (5.0%) | 4 (12.9%) | 1 (5.6%) | 0.21 |

Respondents were allowed to give up to three answers, therefore the total can be higher than the total number of respondents

* P<0.05

P<0.01). Another measure mentioned by 46.6% of the respondents was more telephone consultations instead of face-to-face consultations. A third of the respondents (37.7%) said that a worthwhile change was that they had the impression that women made a better-informed choice about place of birth, which was mentioned more often by community midwives than respondents working in hospital-based care (54.9% vs. 29.5%, P<0.01). Video consultations were a measure that 23.9% of the respondents wanted to keep for future care. A few respondents mentioned innovative organizational structures, like monitoring women at home (for example through cardiotocography) and having online work meetings. Some respondents named other ways of providing personal care, such as providing digital information via online presentations and videos to inform pregnant women, or designing personalized care schedules. Logistic regression analysis showed that video consultations as an innovation were preferred less often by respondents with a work experience shorter than five years, compared with respondents with over twenty years of work experience (OR = 0.05, 95% CI = 0.04–0.61, P = 0.01). Other regression analyses showed no significantly different outcomes for any of the other variables.

**Table 4. Opportunities for the organization of maternity care in the future.**

| | Total (n = 427) | % | A: Community midwife (n = 326) | B: Hospital-based midwife (n = 55) | C: Obstetrican (n = 30) | D: Resident obstetrics (n = 16) | P-value between A and B+C+D |
|---|---|---|---|---|---|---|---|
| A decrease in consultations and ultrasounds when there is no medical indication | 214 | 50.1% | 150 (46.0%) | 41 (74.5%) | 13 (43.3%) | 10 (62.5%) | <0.01* |
| More telephone consultations instead of face-to-face consultations | 199 | 46.6% | 141 (43.3%) | 29 (52.7%) | 19 (63.3%) | 10 (62.5%) | 0.02* |
| Women make better informed choices about place of birth | 161 | 37.7% | 146 (44.8%) | 13 (23.6%) | 1 (3.3%) | 1 (6.3%) | <0.01* |
| Video consultations | 102 | 23.9% | 70 (21.5%) | 14 (25.5%) | 12 (40.0%) | 6 (37.5%) | 0.05* |
| Better collaboration between community-based and hospital-based care | 80 | 18.7% | 65 (19.9%) | 4 (7.3%) | 9 (30.0%) | 2 (12.5%) | 0.31 |
| Fewer people present during labour | 69 | 16.2% | 38 (11.7%) | 22 (40.0%) | 7 (23.3%) | 2 (12.5%) | <0.01* |
| Innovative organizational structures | 50 | 11.7% | 37 (11.3%) | 4 (7.3%) | 6 (20.0%) | 3 (18.8%) | 0.72 |
| Other | 39 | 9.1% | 33 (10.1%) | 5 (9.1%) | 1 (3.3%) | 0 (0.0%) | |
| None | 27 | 6.3% | 24 (7.4%) | 1 (1.8%) | 1 (3.3%) | 1 (6.3%) | 0.16 |

Respondents were allowed to give up to three answers, therefore the total can be higher than the total number of respondents

* P<0.05

## Other results

Some community midwives experienced fewer referrals for women to the hospital for decreased fetal movements (14.8%) (Tables 5 and 6). The main reason respondents gave was that community midwives received fewer calls from women for decreased fetal movements. At the same time, as many as 64.2% of respondents working in hospital-based care experienced fewer consultations for decreased fetal movements.

Nearly all community midwives (93.1%) stated that they experienced no difference in timing of referral during labour compared to the period before COVID-19. Still, of the respondents working in hospital-based care, 42.1% felt women were being referred later than before.

If a woman was referred to the hospital, the majority of community midwives (70.0%) did not experience any delays at the hospital. However, if they did experience a delay, 60.0% attributed this to a problem with the organization of care (capacity problems or triage at the hospital). Only one participant mentioned a lack of ambulances causing the delay.

Regarding the location of labour, 38.5% of community midwives stated they proposed a different location than originally planned by the woman, mainly being the woman's own home. Community midwives working in the East-Netherlands were less likely to suggest a different location for labour (OR = 0.50, 95% CI = 0.26–0.97, P = 0.04). Community midwives working in North-Netherlands were more likely to propose a different location (OR = 3.57, 95% CI = 1.48–8.61, P = 0.01). Of the respondents working in the hospital, 65.3% stated they had the impression that there were more homebirths.

A quarter of the respondents (24.2%) working at the hospital experienced fewer inductions of labour. Very few participants experienced a difference in the number of caesarean sections.

Most respondents working in hospital-based care (85.0%) experienced no capacity problems in their hospital (see S1 and S2 Tables). However, if respondents experienced any capacity problems, this was mostly in the labour rooms, and not in the neonatal unit. There was no significant difference between professions.

**Table 5. Care management by community midwives.**

| | Community midwife (n = 317) | % |
|---|---|---|
| *What was the influence of COVID-19 on referring women for a consultation for decreased fetal movements?* | | |
| No influence | 258 | 81.4% |
| I referred fewer women | 47 | 14.8% |
| I referred more women | 12 | 3.8% |
| *If you referred fewer women, why?* | n = 47 | |
| Fewer women reported decreased fetal movement | 32 | 68.1% |
| Because of fear of women | 8 | 17.0% |
| Because of fear for capacity problems in the hospital | 3 | 6.4% |
| Because I was afraid | 1 | 2.1% |
| Other | 3 | 6.4% |
| *What was the influence of COVID-19 on referral of women during labour?* | | |
| No influence | 295 | 93.1% |
| I referred women later | 18 | 5.7% |
| I referred women earlier | 4 | 1.3% |
| *If referred earlier, why?* | n = 4 | |
| Because of fear for too little capacity of the ambulance | 1 | 25.0% |
| Other | 3 | 75.0% |
| *If referred later, why?* | n = 18 | |
| Because of fear of the woman | 6 | 33.3% |
| Other | 6 | 33.3% |
| Because of fear for capacity problems in the hospital | 4 | 22.2% |
| Because I was afraid | 2 | 11.1% |
| *What was the influence of COVID-19 on the place of birth* | | |
| No influence | 195 | 61.5% |
| I suggested a different place of birth: | 122 | 38.5% |
| • Home | 109 | 89.3% |
| • A different hospital | 7 | 5.7% |
| • In a birth centre | 2 | 1.6% |
| • Community midwife-led hospital birth | 2 | 1.6% |
| • Obstetrican-led care | 2 | 1.6% |
| • Other | 2 | 1.6% |
| *Were there any delays when admitting women to hospital?* | | |
| Yes, there were delays | 51 | 16.1% |
| Neutral | 39 | 12.3% |
| No, there were no delays | 222 | 70.0% |
| Other | 5 | 1.6% |
| *If there was delay, why?* | | |
| Lack of ambulances | 1 | 2.0% |
| Women did not want to be admitted | 9 | 18.4% |
| Capacity problems in the hospital | 12 | 24.5% |
| Due to the triage at the hospital | 17 | 34.7% |
| Other | 10 | 20.4% |

**Table 6. Care management by professionals in hospital-based care.**

|  | Hospital-based care (n = 95) | % |
|---|---|---|
| *What was the influence of Covid-19 on the amount of consultations for decreased fetal movements?* |  |  |
| No influence | 24 | 25.3% |
| There were fewer consults | 61 | 64.2% |
| There were more consults | 4 | 4.2% |
| I do not know | 4 | 4.2% |
| Other | 2 | 2.1% |
| *What was the influence of COVID-19 on the number of women referred for labour at the hospital?* |  |  |
| No influence | 29 | 30.5% |
| Women were referred later | 40 | 42.1% |
| Fewer women were referred | 10 | 10.5% |
| Women were referred earlier | 9 | 9.5% |
| I do not know | 2 | 2.1% |
| Other | 5 | 5.3% |
| *What was the influence of COVID-19 on the place of birth?* |  |  |
| No influence | 33 | 34.7% |
| Birth was more often on a different location: | 62 | 65.3% |
| • Home | 60 | 96.8% |
| • In a birth centre | 1 | 1.6% |
| • Obstetrican-led care | 1 | 1.6% |
| *What was the influence of Covid-19 on the induction of labour?* |  |  |
| No influence | 62 | 65.3% |
| There were fewer inductions of labour | 23 | 24.2% |
| There were more inductions of labour | 8 | 8.4% |
| Other | 2 | 2.1% |

Regarding the safety of healthcare provision, 48.0% of the respondents had the feeling that this had been compromised due to the changes. The majority of respondents (62.7%) attributed this to the decrease in face-to-face consultations.

Of all respondents, 77.0% answered that interprofessional collaboration within the maternity care system was the same or better than before. The main reason was that the respondents experienced better communication and better contact with colleagues (44.7%). If cooperation was experienced as worse, it was mainly due to limited access to the hospital (42.3%).

The majority of respondents (77.7%) stated that specific agreements were made within the maternity care collaboration. Examples mentioned were ensuring that midwifery practices within the same region followed the same rules to prevent women going to practices with more lenient rules, and a central distribution of PPE. The majority of the respondents (79.2%) experienced a clear policy on the use of PPE, with no difference between different professions. More than half of the respondents (64.4%) experienced less job satisfaction.

## Discussion

This study investigated the opinions and experiences of maternity care professionals with the changes in the organization of maternity care in the Netherlands after COVID-19. This study shows that, overall, approximately half of the respondents felt that the safety of maternity care was compromised due to the policy changes. A decrease in consultations was seen as an

important measure, because it reduced provider-patient contact. However, respondents felt that, due to this decrease, pregnant women and partners experienced more insecurity. Fewer women were referred to the hospital for decreased fetal movements. Maternity care professionals felt that women appeared to give more thought to the necessity and safety of ultrasounds and medical interventions. The percentage of homebirths, according to respondents, seemed to have increased. During labour, fewer people were present because only the partner was allowed to attend the birth. A disadvantage of only allowing one person to be present during labour, was that some community midwives were not allowed to give a personal handover after referral to a hospital and could not stay with the woman during labour.

More than half of the respondents stated it was an advantage to have fewer prenatal consultations if they were not strictly medically necessary. Although several respondents stated it was an advantage to have fewer prenatal consultations, others worried about safety being compromised due to fewer face-to-face consultations. Limited research has been done on the minimal number of prenatal consultations. A Cochrane review by Dowswell et al. (2015) analyzed the effect of reduced prenatal consultations in low-risk pregnancies [21]. In the group with reduced prenatal consultations in high-income countries, women had on average eight to twelve prenatal consultations, 2.6 consultations less than the group with care as usual. This did not lead to increased perinatal mortality. There was an increase in preterm births in the reduced visits group, however, results were only marginally statistically significant (risk ratio 1.24, 95% CI 1.01–1.51). There is some evidence that shows that a reduced number of prenatal consultations leads to women being less satisfied with the care given, which may be related to receiving too little non-medical support [21, 22]. The WHO guideline on antenatal care also states that women highly value a positive pregnancy experience, and psychosocial and emotional support [6]. However, since our research was only aimed at the opinion of maternity care professionals, it would be interesting to also investigate how women themselves actually experienced the reduction in prenatal consultations and how this has affected the quality of care they felt they received. Professionals should personalize the number of prenatal consultations to ensure medical and emotional safety is guarded while avoiding unnecessary care [5, 23].

Over the past 30 years, the percentage of homebirths in the Netherlands has decreased from approximately 40% to 13% of all births [24]. The results of our study indicate that professionals feel that there may have been an increase in homebirths during COVID-19; however, clear figures of this potential increase have yet to be released. Several studies show that the outcomes of low-risk births, assisted by a community midwife, are similar at home, in a birth centre or in a hospital [25, 26]. For multiparous women, neonatal outcomes (Apgar scores and NICU admissions) seem to be better for homebirths [27]. Women who plan a home birth have fewer medical interventions [28]. Nevertheless, over the last decade, media reports in the Netherlands have emphasized the potential risks of home birth, leading to fewer women choosing home birth [24]. Participants mentioned that women appeared to be less apprehensive about home birth, and some indicated that women were reluctant to go to a medical facility. This may have resulted in women weighing the advantages and disadvantages of home- versus hospital birth differently. In the United Kingdom home births were restricted during COVID-19 as there was limited access to ambulances [29]. However, in our survey, this was only mentioned by one participant. Therefore, it seems that the availability of ambulances did not contribute to the perceived increase in homebirths during COIVD-19.

More than half of the respondents working in hospital-based care mentioned that they experienced fewer consultations for decreased fetal movements. This was confirmed by 14% of the community midwives. The assessment of fetal movements is important, as a decrease in fetal movements is associated with adverse perinatal outcomes and an increased risk of

caesarean delivery [30]. Fetal movements are assessed better if women are lying down [31], so perhaps if women stayed at home more, they might have had more time to assess fetal movements. On the other hand, women might want to minimize going to the hospital or midwife to prevent getting infected with COVID-19. A study by Linde et al. showed that a possible reason for delay is that women do not want to be a burden to the health care professionals [32]. With an increasing workload in the hospitals caused by COVID-19, this reason may now be even more pertinent. The question remains whether women actually experienced less often decreased fetal movements, or whether they were too reluctant to see a health care professional due to various reasons. When annual national data by Perined (the Dutch National Registry) on perinatal complications will become available, they may give more insight into this [3]. Additionally, our study exploring experiences of women and their partners with maternity care during the COVID-19 pandemic, which is also part of the WAAG study, will give more information on the women's perspective.

Respondents disagreed about the limited number of people allowed to be present during labour. Nearly a quarter thought this was a disadvantage, whereas a third considered this to be an advantage. In addition, more than half of the participants indicated the fact that the community midwife was not allowed to stay was a disadvantage. Previous studies have demonstrated that continuous support of women in labour by a (semi-) professional is beneficial for women's feelings of safety and their feelings about the birth itself; it increases the chance of spontaneous vaginal birth and decreases the chance of interventions during birth [33–35]. A Cochrane review by Bohren et al. (2017) shows that the benefits of continuous support are independent of the relationship of the person providing that support (community midwife, nurse, family, friend or doula) to the woman in labour. Subgroup analysis showed that the only difference was that the presence of a doula was slightly more effective in reducing caesarean sections [34]. However, other studies have shown that a personal handover by the community midwife when women are referred from community-based to hospital-based care has shown to be of advantage for women in labour [33]. On top of that, a Cochrane review on midwife-led continuity of care models, where a (community) midwife is the lead professional throughout pregnancy, labour and the postpartum period, shows that these models are beneficial to both woman and baby [36]. In conclusion, a personal handover and continued attendance of the community midwife during labour is recommended for improving the quality of care in a future crisis situation.

A quarter of respondents stated that they would like to keep the video consultations. Some respondents also mentioned home monitoring for pregnant women, for example for blood pressure or cardiotocography. E-health is currently increasing in all sectors of medicine. Research has been conducted on telemonitoring during pregnancy. An extensive review by van den Heuvel et al. (2018) elaborates on the use of telemonitoring for cardiotocography, which has been found to be effective[37]. Another possibility is that a consultation for decreased fetal movements or post-term pregnancy, including antenatal CTG and ultrasound, is performed by community midwives. Currently, this is piloted and evaluated in the Netherlands. This could provide an additional opportunity for women that are reluctant to go to the hospital because they are afraid of getting infected.

Limited studies on video consultations in maternity care show increased satisfaction among women receiving both video consultations and face-to-face consultations [38], and similar maternal and neonatal outcomes [39], compared with women who receive regular antenatal care. The results of these studies and our study suggest benefits to implementing more video consultations in the future, however, due to the limited studies thus far, more research has to be done to ensure the safety and feasibility of video consultations. Until now it is unknown to what extend the increase in online consultations has an impact on the quality of

care received by women with a social disadvantage due to limitations in internet access and communication skills.

The qualitative follow-up study of maternity care professionals' experiences during COVID-19 which is part of the WAAG study, should yield more depth and background to our findings and the experienced proportionality of the measures taken in the organization of maternity care.

## Strengths and limitations

First, the speed of initiation of the study after the beginning of the COVID-19 pandemic is a strength of this study. Within three months of the first patient being diagnosed with COVID-19, the questionnaire was distributed among maternity care professionals. With a diverse study-group, we were able to develop a comprehensive questionnaire, generating a wealth of information regarding measures taken in the organization of maternity care.

A limitation of this study is the set-up. As we conducted a cross-sectional study, respondents had to remember how they experienced the situation a few months ago. Second, the majority of the respondents reached through social media were community midwives. In the Netherlands, there are more community midwives (N = 2473) compared to hospital-based midwives (N = 967), obstetricans, and residents in obstetrics (combined N = 1408) [40, 41], but the percentage of obstetricans and residents in obstetrics that filled out the questionnaire was fairly small. Perhaps this could have been larger if respondents had been addressed through direct mail.

## Conclusion

Our study shows that maternity care providers have experienced that routine medical care could be safely scaled down. However, psychological and social support during pregnancy are equally important for good quality care, and therefore personalized care should be considered when scaling down routine care. Women having to go to consultations alone during the COVID-19 pandemic was seen as very undesirable by most maternity care professionals, and should therefore be prevented during a next crisis situation.

Equally, the community midwife should be allowed to give a personal handover and stay with the woman for the remainder of the birth, even in times of restricted interpersonal contact.

Video and telephone consultations were seen as improvements, and could therefore in certain cases be alternated with face-to-face consultations.

## Supporting information

**S1 Questionnaire.**
(DOCX)

**S1 Table. Professionals' experiences.**
(DOCX)

**S2 Table. Agreements within maternity care collaboration.**
(DOCX)

## Author Contributions

**Conceptualization:** Eline L. M. van Manen, Martine Hollander, Esther Feijen-de Jong, Ank de Jonge, Corine Verhoeven, Janneke Gitsels.

**Data curation:** Eline L. M. van Manen.

**Formal analysis:** Eline L. M. van Manen, Martine Hollander, Janneke Gitsels.

**Investigation:** Eline L. M. van Manen.

**Methodology:** Eline L. M. van Manen, Martine Hollander, Esther Feijen-de Jong, Ank de Jonge, Corine Verhoeven, Janneke Gitsels.

**Project administration:** Eline L. M. van Manen, Ank de Jonge, Corine Verhoeven.

**Software:** Eline L. M. van Manen.

**Supervision:** Martine Hollander, Esther Feijen-de Jong, Ank de Jonge, Corine Verhoeven, Janneke Gitsels.

**Visualization:** Martine Hollander, Esther Feijen-de Jong, Ank de Jonge, Corine Verhoeven, Janneke Gitsels.

**Writing – original draft:** Eline L. M. van Manen.

**Writing – review & editing:** Martine Hollander, Esther Feijen-de Jong, Ank de Jonge, Corine Verhoeven, Janneke Gitsels.

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
