## [Decision Letter · Decision Letter 0]

18 Mar 2021

PONE-D-21-05490

Experiences of maternity care professionals during the first wave of COVID-19 in a community based maternity care system

PLOS ONE

Dear Dr. van Manen,

Thank you for submitting your manuscript to PLOS ONE. After careful consideration, we feel that it has merit but does not fully meet PLOS ONE’s publication criteria as it currently stands. Therefore, we invite you to submit a revised version of the manuscript that addresses the points raised during the review process.

We look forward to receiving your revised manuscript.

Kind regards,

Hannah Dahlen, RN, RM, BN (Hons), MCommN, PhD FACM

Academic Editor

PLOS ONE

Journal Requirements:

Additional Editor Comments (if provided):

Reviewers' comments:

Reviewer's Responses to Questions

**Comments to the Author**

1. Is the manuscript technically sound, and do the data support the conclusions?

Reviewer #1: Yes

Reviewer #2: Yes

2. Has the statistical analysis been performed appropriately and rigorously? 

Reviewer #1: Yes

Reviewer #2: I Don't Know

3. Have the authors made all data underlying the findings in their manuscript fully available?

Reviewer #1: Yes

Reviewer #2: No

4. Is the manuscript presented in an intelligible fashion and written in standard English?

Reviewer #1: Yes

Reviewer #2: Yes

5. Review Comments to the Author

Reviewer #1: Thank you for the opportunity to review this paper.

The paper is interesting as it looks at a community-based focus. I have a few comments and suggestions.

In the title or abstract it would be helpful to have the country in which the study was conducted. There are literally hundreds of studies now globally about the experiences of care and given countries each had different trajectories of the pandemic, context is essential.

It is hard to see this paper in isolation to the outcomes for women. For example, did attending less for decreased fetal movements make a difference to the rates of stillbirth? I know the woman’s experiences are reported separately but understanding the implications of these changes is difficult. Can the authors report any data that highlights the implications?

I was surprised to see the high numbers of recommended antenatal visits (13). In most countries in high income countries, the recommended number is 7-10 (certainly in Australia this is the case). Could it be that COVID-19 restrictions have just brought the number of antenatal visits down to what is more evidence-based anyway?

I was also surprised that no human ethical approval was required. Was this the same for the women’s survey?

The model of care in The Netherlands was described. Are women receiving community-based care more likely to have continuity of carer, that is a known midwife, than in the hospital setting? If so, this probably made a difference and it could be the continuity that explains the differences rather than the place of care.

How was social disadvantage addressed during the pandemic? For example, when care shifted to the phone or online, how did women without ready access to these modalities receive care? How did language needs get addressed?

Reviewer #2: Thank you for the opportunity to review this interesting and relevant manuscript. I have found the manuscript to be well written and I have a few comments for review.

1. Methods / Study design - It is written that ethical approval was not deemed necessary by the medical ethics committee. I think further information is needed regarding this. Was it because this was part of a larger study that had an umbrella approved ethics application and that this part didn't need a separate ethics?

2. Analysis - line 152 - you mention 'free text fields were either recorded into existing categories or new categories. I think further clarification is required here to add these free text fields were from the 'other' option given in a list and then the answer from the 'other' option was recorded as an existing or new category. When I first read this I thought you had open ended questions and then you were doing a content analysis until I viewed the survey questions at the end.

3. Discussion - line 345 - What data are you referring to when you mention the 'data on perinatal complications', is this data you have collected in the wider study or national perinatal data?

4. Lines 390-382 - Is the qualitative follow up study you mention part of the wider study? This needs clarification.

5. Strengths and limitations line 385-386 - This sentence could be reworded to be clearer. A suggestion is: The speed of initiation of the study after the beginning of the COVID-19 pandemic is a strength of this study.

6. PLOS authors have the option to publish the peer review history of their article (what does this mean?). If published, this will include your full peer review and any attached files.

Reviewer #1: **Yes: **Caroline Homer

Reviewer #2: **Yes: **Hazel Keedle

---

## [Author Response · Author response to Decision Letter 0]

15 Apr 2021

Reviewer #1: Thank you for the opportunity to review this paper.

The paper is interesting as it looks at a community-based focus. I have a few comments and suggestions.

In the title or abstract it would be helpful to have the country in which the study was conducted. There are literally hundreds of studies now globally about the experiences of care and given countries each had different trajectories of the pandemic, context is essential.

In both the abstract, the title and the short title, the word ‘Dutch’ has been added (page 1, line 1 and 3, and page 2, line 35).

It is hard to see this paper in isolation to the outcomes for women. For example, did attending less for decreased fetal movements make a difference to the rates of stillbirth? I know the woman’s experiences are reported separately but understanding the implications of these changes is difficult. Can the authors report any data that highlights the implications?

It is indeed very relevant to compare the outcomes of our study to perinatal outcomes. However, national annual data by Perined (the Dutch National Registry) are not yet available. In our study perinatal and maternal outcomes have therefore not been analysed. However, there is another study from our group exploring the experiences of women and their partners with maternity care during this episode that is almost ready for publication, and will be a suitable companion to this study. We refer to this in our manuscript (page 17, line 349-353).

I was surprised to see the high numbers of recommended antenatal visits (13). In most countries in high income countries, the recommended number is 7-10 (certainly in Australia this is the case). Could it be that COVID-19 restrictions have just brought the number of antenatal visits down to what is more evidence-based anyway?

You are correct that we have minimized the number of prenatal visits to what is the evidence based advice based on medical outcomes, in line with recommendations of the World Health Organisation (WHO) and international guidelines [1, 2]. The additional visits are based on the guideline on prenatal care from the Dutch Organisation of Midwives and are supported by evidence that women appreciate regular contact for support and information during pregnancy [3]. This was considered less essential during the crisis. This has been clarified further in the text (page 2, line 76-80). 

I was also surprised that no human ethical approval was required. Was this the same for the women’s survey?

The ethics committee found that no ethical approval was necessary because participants received no medical interventions, and the emotional burden of the questions was not considered to be so severe that approval of a medical ethical committee was warranted (page 4, line 128-130). This was the same for both the women’s survey and the interviews. 

The model of care in The Netherlands was described. Are women receiving community-based care more likely to have continuity of carer, that is a known midwife, than in the hospital setting? If so, this probably made a difference and it could be the continuity that explains the differences rather than the place of care.

The reviewer is correct in thinking that women in community-based care are more likely to know the midwife who attends their birth. However, the primary care midwife will attend to women in labour regardless of their chosen place of birth and therefore their choice has no influence on continuity of care. We have clarified this (page 2, line 67). 

How was social disadvantage addressed during the pandemic? For example, when care shifted to the phone or online, how did women without ready access to these modalities receive care? How did language needs get addressed?

This is a very pertinent question, however, not one that has an easy answer. The best we can say is that almost all women in the Netherlands have access to a mobile phone, therefore, phone consultations have been accessible to (almost) all women. We do believe that it is possible that quality of care for disadvantaged women has been reduced during the initial stages of the pandemic. However, there are no data on this as yet. We have added a comment clarifying this is in the manuscript (page 18, line 386-388).

Reviewer #2: Thank you for the opportunity to review this interesting and relevant manuscript. I have found the manuscript to be well written and I have a few comments for review.

1. Methods / Study design - It is written that ethical approval was not deemed necessary by the medical ethics committee. I think further information is needed regarding this. Was it because this was part of a larger study that had an umbrella approved ethics application and that this part didn't need a separate ethics?

We refer to our comment on this matter to a question from reviewer 1.

2. Analysis - line 152 - you mention 'free text fields were either recorded into existing categories or new categories. I think further clarification is required here to add these free text fields were from the 'other' option given in a list and then the answer from the 'other' option was recorded as an existing or new category. When I first read this I thought you had open ended questions and then you were doing a content analysis until I viewed the survey questions at the end.

In the manuscript we now mentioned that free text fields were marked as other (page 4, line 157).

3. Discussion - line 345 - What data are you referring to when you mention the 'data on perinatal complications', is this data you have collected in the wider study or national perinatal data?

Throughout the Netherlands national data on perinatal complications are routinely recorded and this is annually analyzed, through a registry called Perined. The data of 2020 are not yet available. We clarified this in the manuscript (page 17, line 349). 

4. Lines 390-382 - Is the qualitative follow up study you mention part of the wider study? This needs clarification.

This qualitative follow up study is indeed part of the wider study, the WAAG study. This has now been clarified in the text (page 18, line 389-390). 

5. Strengths and limitations line 385-386 - This sentence could be reworded to be clearer. A suggestion is: The speed of initiation of the study after the beginning of the COVID-19 pandemic is a strength of this study.

Thank you for this suggestion. We changed the manuscript according to this suggestion (page 18, line 394-395). 

Changes made to reference list

References to the guidelines from the World Health Organization (WHO) (reference 6) and international guidelines (reference 7) were added.

We specified reference 22, now the specific chapter that has been used as a reference has been added to the reference. 

References

1. NICE. Antenatal care for uncomplicated pregnancies: schedule of appointments. 2020.

2. WHO. WHO recommendations on antenatal care for a positive pregnancy experience: World Health Organization; 2016.

3. KNOV. KNOV-standaard: Prenatale verloskundige begeleiding. 2008.

---

## [Decision Letter · Decision Letter 1]

21 May 2021

Experiences of Dutch maternity care professionals during the first wave of COVID-19 in a community based maternity care system

PONE-D-21-05490R1

Dear Dr. van Manen,

We’re pleased to inform you that your manuscript has been judged scientifically suitable for publication and will be formally accepted for publication once it meets all outstanding technical requirements.

Kind regards,

Hannah Dahlen, RN, RM, BN (Hons), MCommN, PhD FACM

Academic Editor

PLOS ONE

Additional Editor Comments (optional):

Reviewers' comments:

Reviewer's Responses to Questions

**Comments to the Author**

1. If the authors have adequately addressed your comments raised in a previous round of review and you feel that this manuscript is now acceptable for publication, you may indicate that here to bypass the “Comments to the Author” section, enter your conflict of interest statement in the “Confidential to Editor” section, and submit your "Accept" recommendation.

Reviewer #1: All comments have been addressed

Reviewer #2: All comments have been addressed

2. Is the manuscript technically sound, and do the data support the conclusions?

Reviewer #1: Yes

Reviewer #2: Yes

3. Has the statistical analysis been performed appropriately and rigorously? 

Reviewer #1: Yes

Reviewer #2: Yes

4. Have the authors made all data underlying the findings in their manuscript fully available?

Reviewer #1: Yes

Reviewer #2: Yes

5. Is the manuscript presented in an intelligible fashion and written in standard English?

Reviewer #1: Yes

Reviewer #2: Yes

6. Review Comments to the Author

Reviewer #1: Thank you for completing the changes. I am still concerned about the lack of ethical approval but I guess that was out of your hands.

Reviewer #2: Thank you for addressing the comments made in the previous review and I feel this is now acceptable for publication.

7. PLOS authors have the option to publish the peer review history of their article (what does this mean?). If published, this will include your full peer review and any attached files.

Reviewer #1: No

Reviewer #2: No

---

## [Editor Report · Acceptance letter]

8 Jun 2021

PONE-D-21-05490R1 

Experiences of Dutch maternity care professionals during the first wave of COVID-19 in a community based maternity care system 

Dear Dr. van Manen:

I'm pleased to inform you that your manuscript has been deemed suitable for publication in PLOS ONE. Congratulations! Your manuscript is now with our production department. 

Kind regards, 

on behalf of

Dr. Hannah Dahlen 

Academic Editor

PLOS ONE